# The familiarity effect of Chinese stroke stimulus and imagery on contextual integration: Evidence from ERP correlates

**Sam Chi Chung Chan**[1☯*], **Tom Chun Wai Tsoi**[1,2☯]

**1** Department of Rehabilitation Sciences, Applied Cognitive Neuroscience Laboratory, The Hong Kong Polytechnic University, Hong Kong Special Administrative Region, China, **2** Sau Po Centre on Ageing, The University of Hong Kong, Hong Kong Special Administrative Region, China

☯ These authors contributed equally to this work.
* samcc.chan@polyu.edu.hk

## Abstract

The neural process of contextual integration has been examined through the phenomenology of semantic incongruence of words. The present study investigated whether the effort of contextual integration would be heightened by the increased demand of selective attention and attention orientation to unfamiliar Chinese stroke style and sequence. It also examined whether visual imagery of unfamiliar stroke style and sequence would mitigate the effort of contextual integration of unfamiliar Chinese stroke. Nineteen participants take part in two cognitive tasks: (a) imagery of Chinese strokes and (b) detection of Chinese familiar and unfamiliar stroke style. An electroencephalogram was concurrently recorded for the analysis of event-related potential (ERP). Results revealed significant differences in attention orientation and effort of contextual integration between familiar and unfamiliar Chinese strokes, as indicated by larger amplitudes of N160 (100–200 ms) & P200 (260–380 ms) components. Furthermore, a larger amplitude of N400 (300–500 ms) component, signifying the neural process of integrating external from the context, was obtained when individuals viewed unfamiliar Chinese strokes. These findings suggest a cognitive effort was needed to process unfamiliar Chinese stimuli, followed by a greater mental effort required for contextual integration of the unfamiliar stimuli. Furthermore, top-down control of visual imagery would facilitate the process of contextual integration via generating internal representation. This finding provides a new insight that the effort expended in contextual integration may be associated with both attentional control and the generation of internal representation from long-term storage across visual stimuli with varying levels of stimulus familiarity. In summary, our study provided insights into the cognitive mechanisms underlying attentional control, contextual integration, and the role of visual imagery in the processing of stimuli with different levels of familiarity. Furthermore, it suggested the potential utility of the N400 component as a biomarker for assessing attention control and memory retrieval functions.

**Data availability statement:** All relevant data are within the Supporting Information files.

**Funding:** The author(s) received no specific funding for this work.

**Competing interests:** The authors have declared that no competing interests exist.

## 1. Introduction

Our brain constantly receives and attends to sensory information from the external context and shifts attention internally to retrieve the relevant internal representation from long-term memory. This elicits semantic processing of internal representations [1,2], that allows us to interpret the meaning of the stimulus and make sense of incoming bottom-up stimuli from the surrounding context [3]. Recent studies showed that the brain mentally exerts an effort to internalize an incoming representation called contextual integration [1,4–7]. This is involved with the continuous neural processes of integrating external, or "contextual", sensory stimuli into the long-term internal representation or the network of the "world knowledge" in our brain. When an incoming novel representation does not align with internalized long-term storage, individuals are expected to exert additional mental effort to incorporate it into the internal representation, "reshaping" their world knowledge. This study investigated how attention control of stimuli with different levels of familiarity mediates the effect of mental effort of contextual integration.

Previous studies examined the neural process of contextual integration through studying the phenomenon of semantic incongruence of lexicon stimuli [8–10]. For example, when one reads a word with a string of a learned sequence of alphabets "a-p-p-l-e", we know it refers to a type of fruit as we have learned this word formed by these five alphabets and it is associated with a round fruit in red color. In contrast, when a novel word with an unfamiliar sequence of alphabets, for instance, "a-p-r-i-c-o-t", which does not concur with any long-term representation, the neural process of semantic incongruence would be induced. The processing of a novel string of alphabets necessitates heightened attention and cognitive effort to extract semantic meaning based on existing knowledge. Later studies revealed that neural processes of contextual integration might be associated with semantic incongruence in different familiarity levels of visual stimuli other than lexicon-related ones, such as faces [11–13], visual scenes [14–16], and traffic sign identification [17]. This study aimed to investigate if the neural processes of contextual integration would be mediated by non-semantic incongruence when viewing and recognizing unfamiliar sub-character Chinese stimuli under improper stroke sequence. It is proposed that viewing unfamiliar Chinese stimuli, which have higher incongruency with existing long-term storage, would require higher demand of contextual integration to reshape the long-term representation.

Chinese characters in regular style (familiar form comparable to the printed form for English alphabets) are written with a specific sequence of basic strokes (analogous to spelling in English). The basic rules of stroke sequence generally go from top to bottom and left to right. For example, the character "六" (meaning "six") is written in a proper sequence of "丶", "一", "丿" and "丶". It is postulated that a word with an improper sequence which results in an unfamiliar sub-character form and deviates from how strokes typically written, would induce an incongruency between the incoming unfamiliar form with the long-term or stored representations of the character under a proper sequence. Based on previous studies studying the forms of Chinese

character strokes with different familiarity levels [16,18,19], the stroke combinations that are unfamiliar to the viewers compared with familiar stroke combinations in the long-term storage, would induce high attention and cause more effort to encode and store the novel character forms. Furthermore, based on the previous studies examine the process of contextual integration [14,20–22], it is assumed that unfamiliar figures would require additional attention effort to the incongruent details in the figures and the internal representations would also demand higher mental effort to integrate the new figures into the long-term representation. It is hypothesized that unfamiliar stroke forms due to improper stroke sequence would similarly require higher attention and higher demand to integrate. Yet, there is little understanding of how the mental effort of attention of contextual integration would be mediated by sub-characters with different levels of familiarity.

## 1.1. External and internal attention orientation in relation to contextual integration

The cognitive processes of selective attention and external-to-internal attention orientation are stipulated to be related to contextual integration processing. Selective attention refers to attention control to process the relevant information in the visual stimuli, and external-to-internal attention orientation direct attention from the external context to the processed internal representation for further integration to the existing storage, A frontally distributed N160 component (100–200 ms post-stimulus onset) has been revealed to be associated with orienting attention to external sensory stimuli, i.e., selective attention [23–25] and selective attention on incoming visual representations [21,26–39] (Katus et al., 2012; Legrain et al., 2013; Liu et al., 2014; Ohara et al., 2006; Peterson et al., 2012). Other studies found that an increased amplitude of the P200 component (260–380 ms post-stimulus onset) at fronto-central sites is associated with the additional effort of external-to-internal attention orientation [30]. Furthermore, previous studies showed that unfamiliar stimuli have been shown to require higher attentional control as reflected by increased amplitude of the N160 component [31–33]. The studies which investigated attentional control on orthographical stimuli showed that higher attention orientation to external stimuli to internal representations was required to process unfamiliar pseudo-letters with larger P200 amplitude [27,34–36]. Based on these findings, contextual integration would require effective top-down control of attention selection on external stimuli, followed by attention orientation from the external environment to the internal context. It is speculated that unfamiliar stimuli demand greater selective attention and orientation processes.

## 1.2. Effect of stimulus familiarity on contextual integration reflected by N400

A negative-going N400 (300–500 ms post-stimulus onset) component at frontal sites has been found to be associated with the semantic processing of orthographic stimuli in earlier linguistic studies and could be regarded as associated with the mental effort of contextual integration [4]. A larger amplitude of the N400 component will result if one views an unexpected appearance of a word in a sentence context (for example "He liked drinking coffee with sugar and lemon", where "lemon" is semantically unexpected compared with the expected counterpart "milk"). It was theorized in earlier studies that the frontal N400 component may reflect additional neural activations for processing the incoming incongruent stimulus (i.e., "coffee") in a context [37–39]. Subsequent studies revealed that a larger amplitude of the N400 component was elicited when processing stimuli did not match the context (visual scenes [14,15]; faces [11,12]; traffic sign identification [17]. It was suggested that the N400 component could be associated with contextual integration, which represented mental discrepancy or incongruence between bottom-up external stimulus and the internal representation in the long-term storage such that additional mental effort would be required to internalize or integrate the new "concept" from the external context [4,40]. If the degree of unfamiliarity with stimulus increases, the mental effort for semantic incongruence in the process of contextual integration would be augmented. Recent studies examined the incongruence effect of the N400 component of orthographical components of Chinese words, i.e., radicals or strokes [41,42]. It is expected that, when such an internal representation of a Chinese stroke form or sequence is not well established in the brain network, a stronger mental effort, indicated by a larger N400 amplitude, for contextual integration mediated by enhanced attention control would

also be elicited. In the study conducted by Wang & Dong [42], the Chinese-speaking participants were asked to view radical-omitted and stroke-missing characters. The results showed that the stroke-omitted characters would elicit not only a larger orthographic-related attention (P200 component) but also a larger mental effort of contextual integration (N400 component) when compared with familiar radical characters. The orthographic-related attention reflects the mapping of pre-lexical orthographic representations on the whole word [43]. It is explained that the missing strokes in a character are more unfamiliar to Chinese-speaking participants based on long-term memory storage. Therefore, even an unfamiliar stroke form or sequence may require enhanced attentional control and additional mental effort to integrate the stimulus into a long-term internal representation and in turn reshape the internal representation as indicated by the augmented amplitude of the N400 [1,4,14,43].

### 1.3. Effect of visual imagery on contextual integration indicated by N400

Previous studies have provided evidence that priming stimuli with contextual information can lead to a reduction in the mental effort required for contextual integration, indicated by a decrease in the amplitude of the N400 component [4,44]. In addition, visual imagery, which involves the generation of internal representation without external sensory output, is speculated to activate top-down anticipation and long-term storage. This activation, in turn, is speculated to lower the mental effort required for contextual integration [45–51] However, it is still unclear whether maintaining internal representation of an unfamiliar stroke-level stimuli, which carry no explicit semantics, through visual imagery would still generate a relevant top-down representation from the existing storage to facilitate contextual integration. In this study, it is theorized that a top-down anticipation of the subsequent stroke sequence through visual imagery would reduce the incongruence effect and thus the effort of contextual integration. In terms of the neural network, the findings of this study would give a clear picture on the linkage between different levels of attention control indicated by N160 and P200 components mediated by different levels stimulus familiarity [4,48,52] and the effort of contextual integration and long-term storage of internal visual representation signified by the N400 component [45,46,48–51].

The objectives of this ERP study were to (1) examine how bottom-up selective attention and attention orientation interfered by two levels of stimulus familiarity, reflected by the N160 & P200 component; (2) examine if top-down internal representation would mediate the process of contextual integration by mental generation of visual images with two levels of familiarity, reflected by N400 component. It is hypothesized that viewing Chinese orthographical units or strokes in the unfamiliar style would result in larger N160 & P200 amplitude (attention orientation), and larger N400 amplitude (contextual integration) when compared with those in the familiar counterpart. Furthermore, attention orientation indicated by the N160 and P200 components and the mental effort of contextual integration indicated by the N400 component would be reduced when generating visual imagery of Chinese strokes. The findings would shed light on how contextual integration is related to the long-term memory network. They would further lead to a better understanding of different levels of attention demand and top-down control would mediate the neural processes of contextual integration and long-term memory network.

## 2. Materials and methods

### 2.1. Participants

Nineteen right-handed cognitively intact adults (9 female) recruited and participated voluntarily in the study in the period between 1 March 2017 and 30 April 2018. The mean age of the participants was obtained to be 22.4 years (SD = 2.8). The inclusion criteria were: (1) at the age of 18 or above (2) having secondary education or above; (2) being native Chinese and being able to read Chinese; (3) not having prior formal training in Chinese calligraphy in semi-cursive style, and (4) having normal or corrected-to-normal vision; (4) obtaining at least 28 points in the Hong Kong version of Montreal Cognitive Assessment Hong Kong version (MoCA) [53]. Those who were diagnosed to have any neurological (e.g. Parkinson's disease and stroke) or psychiatric (e.g., depression) disorders were excluded from the study. All of the participants fulfilled the inclusion and exclusion criteria. They all had completed their undergraduate education and were right-handed. The

mean MoCA score was found to be 29.3 (SD = 0.849). This study was approved by the Hong Kong Polytechnic University Human Ethics Committee. The volunteering participants provided written informed consent by signing the consent form after the study procedure stated in an information sheet was explained by the researcher.

## 2.2. Sample size estimation

In an experiment examining the N200 & N400 effect on Chinese characters, the calculated effect size was found to be 0.23 (Wang & Dong, 2013). This indicates a moderate impact on the processing of these characters. To account for a model of 2 cognitive tasks × 2 stimulus types, sample size estimates suggest that this design would be sufficient to compare performance across the four conditions at α = .05 and power = 0.8. The expected correlation among repeated measures was set at 0.5. Consequently, a total of 15 participants are required for the study.

## 2.3. Experimental stimuli

Two types of Chinese strokes with different levels of familiarity were used: semi-cursive style (name of font: *Shinsu* 行書) and regular (name of the font: *SimHei* 黑體) styles which were chosen to represent unfamiliar and familiar stimuli, respectively. By utilizing these distinct stroke styles, we aimed to assess the participants' responses towards strokes with varying levels of familiarity. Chinese characters with several strokes ranging between 6 and 12 were selected. The character complexity of all selected characters was up to primary six. Each selected character in white was represented as a whole (2°·2.5° visual angle) or in partial form on a black background. All characters chosen were in a left-right structure.

## 2.4. Experimental paradigm

The experiment paradigm design was adopted from the one used in the study conducted by Qiu and colleagues (2010). Each participant was required to be engaged in an experiment with two cognitive tasks: (a) imagery of Chinese strokes and (b) detection of Chinese strokes. Each task was involved with two styles of Chinese strokes: semi-cursive (unfamiliar) styles (行書) and regular (familiar) styles (黑體). In the imagery task, each participant was first shown with a whole Chinese word for priming purposes (1200 ms) before a black blank screen appeared. Each stroke of a Chinese character was presented for 800 ms before another stroke was presented cumulatively in proper writing sequence. This cumulative presentation allowed for the gradual formation of the Chinese character to be observed by participants, simulating the natural writing process. After each stroke appeared, the participant was required to generate mentally and rehearse the image of the stroke which is about to appear according to the proper sequence. He or she was randomly presented with two isolated strokes in each trial and was required to determine which stroke present was the same as the image being rehearsed by pressing a button as quickly as possible within 1600 ms. In the detection task, each participant was also shown a whole Chinese word at the beginning of each trial (for 1200 ms). A series of Chinese strokes was presented cumulatively in proper or improper sequence. A question mark was shown randomly in the series. The timing of the response screen was kept consistent across the detection and imagery tasks. This means that participants were given the same amount of time to respond in both tasks. The participant was required to view and judge if the appearance of each new stroke was consistent with the proper sequence by pressing a button with his or her index finger as quickly as possible within 1600 ms. The position of the correct stroke was counterbalanced. This means that the placement of the correct stroke was alternated across different trials or participants to ensure that it did not consistently appear in the same position. The stimulus presentation timing was the same for both semi-cursive and regular styles. For each type of cognitive task with a specific type of stimulus (i.e., semi-cursive styles and regular styles), there were a total of 4 blocks with 16 trials in each block. A total of 32 Chinese characters were included as stimuli, with each character appearing twice, randomly distributed across the 4 blocks in both regular and semi-cursive styles. The block order in our experiment was counterbalanced and randomized. Breaks were given between blocks whenever the participant needed them. Fig 1 illustrates the schematic representation of the experimental paradigm for semi-cursive and regular strokes.

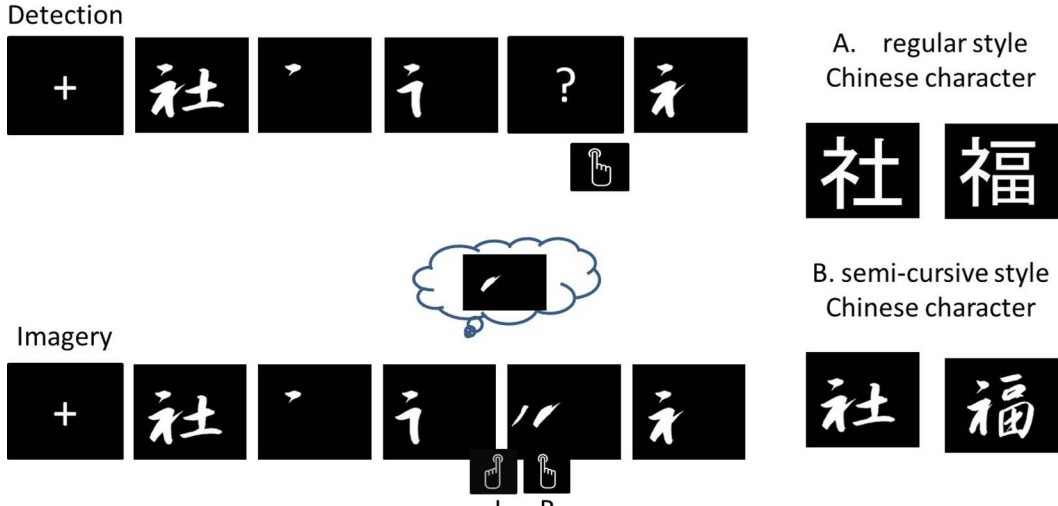

**Fig 1. The experimental paradigm employed in this study involved two tasks: A Perception/Detection task and an Imagery task.** In the Perception/Detection task, participants were instructed to respond by clicking either the right or wrong button when a question mark appeared. On the other hand, in the Imagery task, participants were instructed to indicate the position of the stroke stimulus by responding with either the left or right button.

## 2.5. Experiment procedure

Each participant, who fulfilled the inclusion and exclusion criteria, was asked to have an informed consent signed after the experimenter explained the nature and purpose of the study. Before the experiment, a training session was provided to each participant to understand the two cognitive tasks with two types of Chinese stroke styles. The participant was trained to write selected Chinese strokes with a calligraphy brush to get preliminarily familiarized with the unfamiliar Chinese stroke style. The participant was required to participate in two blocks of experiment trials with both types of cognitive tasks and Chinese stroke styles for them to get familiar with both types of cognitive tasks with both types of stroke stimuli. A minimal accuracy of 70% was needed before the experiment commenced with concurrent EEG signal recording.

During the experiment, the participant was seated comfortably in an electrically shielded and sound-proof chamber and was given breaks periodically to avoid mental fatigue. Each participant was positioned in front of a computer-synchronized display screen with an eye-to-screen distance of 60–65 cm. The viewing angle was set at 5 degrees. His or her right arm was well supported so that the right index finger was positioned on the designated button of a keypad connected to a computer for making responses as required by the experiment.

## 2.6. Neurophysiological data acquisition and preprocessing

An outlier analysis on the neurophysiological data based on the ±3 standard deviations was conducted. No participants were identified as outliers and were included in the analysis. The electroencephalogram (EEG) was recorded concurrently with the behavioral data using 64 electrodes (90 mm Ag/AgCl sintered) mounted on a Quikcap via CURRY Scan 7 Neuroimaging Suite (NeuroScan Labs, Sterling, VA). A head-box of the SynAmps2 Digital DC EEG Amplifier was used to amplify the EEG signals. The montage was referenced to the left and right bony mastoid processes, and the ground electrode was located on the forehead in front of the vertex electrode (Cz). The vertical and horizontal electrooculograms (EOGs) due to blinks and eye saccades were recorded by two separate pairs of electrodes at 2 centimeters above and below the left eye and the outer canthus of each eye, respectively. All electrode impedances were maintained at or below 5kΩ

by injecting conductance gel at each electrode. The timing and presentation of presented stimuli and EEG signals were synchronized by the stimulus presentation program STIM2 (NeuroScan Labs, Sterling, VA).

Offline pre-processing of EEG data was conducted using CURRY Scan 7 software (NeuroScan Labs, Sterling, VA). It began with re-referencing between two mastoid process electrodes, band pass filtering with 0.1–30 Hz, ocular artifact reduction. EEG data between 200 ms pre-stimulus onset to 800 ms post-stimulus was epoched followed by baseline correction. Epochs were extracted from trials in which each stroke of a Chinese character was presented in the proper writing sequence, with a duration of 800 milliseconds for each stroke before another stroke was presented cumulatively. By presenting the strokes in the correct sequence and providing sufficient time for observation, the participants' neural responses were captured, enabling the examination of how familiarity and visual imagery influenced the processing and integration of the strokes. Those muscle or movement artifacts with amplitudes larger than 100μV were rejected. Independent component analysis was applied to the selected epochs using the EEGLAB software (MATLAB platform) to confirm the onset and offset of the ERP component in questions, i.e., N160 (60–130 ms), P200 (130–220 ms) and N400 (250–380 ms) components. The accepted epochs were then averaged according to cognitive (imagery vs detection) tasks and stimulus (semi-cursive vs regular) types.

## 2.7. Statistical analyses

To achieve the first and second objectives, three-way repeated measures ANOVA with a model of 2 cognitive tasks (imagery vs detection) · 2 familiarity levels (semi-cursive and regular Chinese strokes) · 3 midline electrode sites (Fz, Cz, & Pz) were built separately for the amplitudes of N160, P200 and N400 components to examine the main and interaction effects between various parameters. Midline electrode sites (Fz, Cz, and Pz) were chosen for the analysis because the cognitive processes associated with the three ERP components correspond to these cortical regions. Previous studies examining similar cognitive processes also included these midline sites in their analyses [4,15]. Greenhouse-Geisser correction adjustment was applied when the sphericity assumption was violated. Post-hoc contrast tests were conducted for any significant interaction effects with Bonferroni's adjustments set at a p-value of 0.01.

## 3. Results

### 3.1. Behavioral measurements

In this study, the reaction time and accuracy rate of making the response in each trial of the detection condition and the imagery condition with regular or semi-cursive strokes were recorded. The mean and standard deviation (SD) of these behavioral performance in the four experimental conditions was summarized in Table 1.

The average reaction time for the detection condition using regular style recorded a mean of 673.04 ms (SD = 170.42 ms). In contrast, the average reaction time for the detection condition using semi-cursive style Chinese was 700.83 ms

**Table 1. Accuracy and reaction time in detection and imagery conditions of the experience.**

| Accuracy | Detection | | Imagery | |
| --- | --- | --- | --- | --- |
| | Regular | Semi-cursive | Regular | Semi-cursive |
| Mean | 0.92 | 0.91 | 0.84 | 0.72 |
| SD | 0.07 | 0.07 | 0.09 | 0.07 |
| Reaction Time | Detection | | Imagery | |
| | Regular | Semi-cursive | Regular | Semi-cursive |
| Mean | 673.04 | 700.83 | 877.34 | 978.92 |
| SD | 170.42 | 161.72 | 135.31 | 149.21 |

Note. Accuracy and reaction time were recorded; SD stands for the standard deviation

(SD = 161.72 ms). For the imagery condition, the average reaction time with regular style was 877.34 ms (SD = 135.31 ms), while the average reaction time for the imagery condition using semi-cursive style was 978.92 ms (SD = 149.21 ms).

Regarding accuracy, the average accuracy for the detection condition with regular style showed a mean of 0.92 (SD = 0.07), while the average accuracy for the detection condition with semi-cursive style achieved a mean of 0.91 (SD = 0.07). The average accuracy for the imagery condition with regular style recorded a mean of 0.84 (SD = 0.09), whereas the average accuracy for the imagery condition with semi-cursive style had a lower mean of 0.72 (SD = 0.07).

The results showed that reaction times were comparable across both styles in the detection condition. However, the reaction time for the semi-cursive style in the imagery condition was longer than that for the regular style. Additionally, while performance in the detection tasks was similar for both styles, the semi-cursive style demonstrated lower accuracy in the imagery tasks compared to the regular style. A minimal accuracy of 70% was required before the experiment commenced with EEG signal recording. Although the behavioral measurements did not fully align with the EEG data, the fact that all participants met the accuracy criterion ensured that the performance levels were reliably represented.

### 3.2. The N160 component – External attention control

Repeated measures ANOVA showed a significant cognitive tasks × stimulus styles interaction effect of the amplitude of N160 component at midline sites [Fz: $F_{1,18}$ = 13.71, p = 0.002, $\eta^2$ = 0.432; Cz: $F_{1,18}$ = 13.12, p = 0.002, $\eta^2$ = 0.422]. The post-hoc analysis revealed significantly more negative-going N160 at the midline sites when the participants performed visual imagery of unfamiliar semi-cursive strokes compared with detection of the same type of stimulus [$t_{18}$ = 3.46, p = 0.003 at Fz & $t_{18}$ = 4.03, p < 0.001 at Cz]. In contrast, imagery of familiar regular strokes did not reveal any significant task effect of N160 amplitude at the selected midline sites [$t_{18}$ = −0.46, p = 0.165 at Fz & $t_{18}$ = 0.46, p = 0.65].

### 3.3. The P200 component – External-to-internal attention orientation

Repeated measures ANOVA showed a significant cognitive tasks × stimulus styles interaction effect of the amplitude of the P200 component at midline sites [Fz: $F_{1,18}$ = 7.16, p = 0.015, $\eta^2$ = 0.285; Cz: $F_{1,18}$ = 6.05, p = 0.024, $\eta^2$ = 0.251]. Post-hoc analysis revealed significantly less positive-going P200 amplitude at the midline sites when the participants performed imagery of semi-cursive strokes compared with the detection task of the same stimulus [$t_{18}$ = 3.32, p = 0.004 at Fz & $t_{18}$ = 3.84, p = 0.001 at Cz]. On the other hand, no significant effect was revealed in the amplitude of the P200 component when the participants performed the imagery and detection tasks of regular strokes in the selected midline sites [$t_{18}$ = 0.33, p = 0.747 at Fz & $t_{18}$ = 1.21, p = 0.244 at Cz].

### 3.4. The N400 component – Contextual integration effect

Repeated measures ANOVA showed a significant cognitive tasks × stimulus styles interaction effect in the amplitude of the N400 at the midline sites [Cz: $F_{1,18}$ = 5.00, p = 0.038, $\eta^2$ = 0.217; Pz: $F_{1,18}$ = 4.75, p = 0.043, $\eta^2$ = 0.209]. Post-hoc analysis revealed that N400 amplitude was larger for the imagery of unfamiliar stimuli at midline sites [$t_{18}$ = 4.48, p < 0.001 at Fz, $t_{18}$ = 5.79, p < 0.001 at Cz & $t_{18}$ = 4.34, p < 0.001]. Imagery of familiar regular strokes revealed no significant difference in N400 amplitude across midline sites [$t_{18}$ = 0.29, p = 0.777 at Fz, $t_{18}$ = 1.38, p = 0.184 at Cz & $t_{18}$ = 1.15, p = 0.267] compared with detection. Fig 2 illustrates the difference between the two tasks across Fz & Cz sites while the last arrow indicates the N160, P200 and N400 components.

## 4. Discussion

Through the context of viewing and rehearsing Chinese orthographical strokes, this study investigated the effect of attention and the top-down image generation of visual representation of stimulus with different level familiarities that could mediate the process of contextual integration. The mental effort for the selective attention of the unfamiliar stimuli was

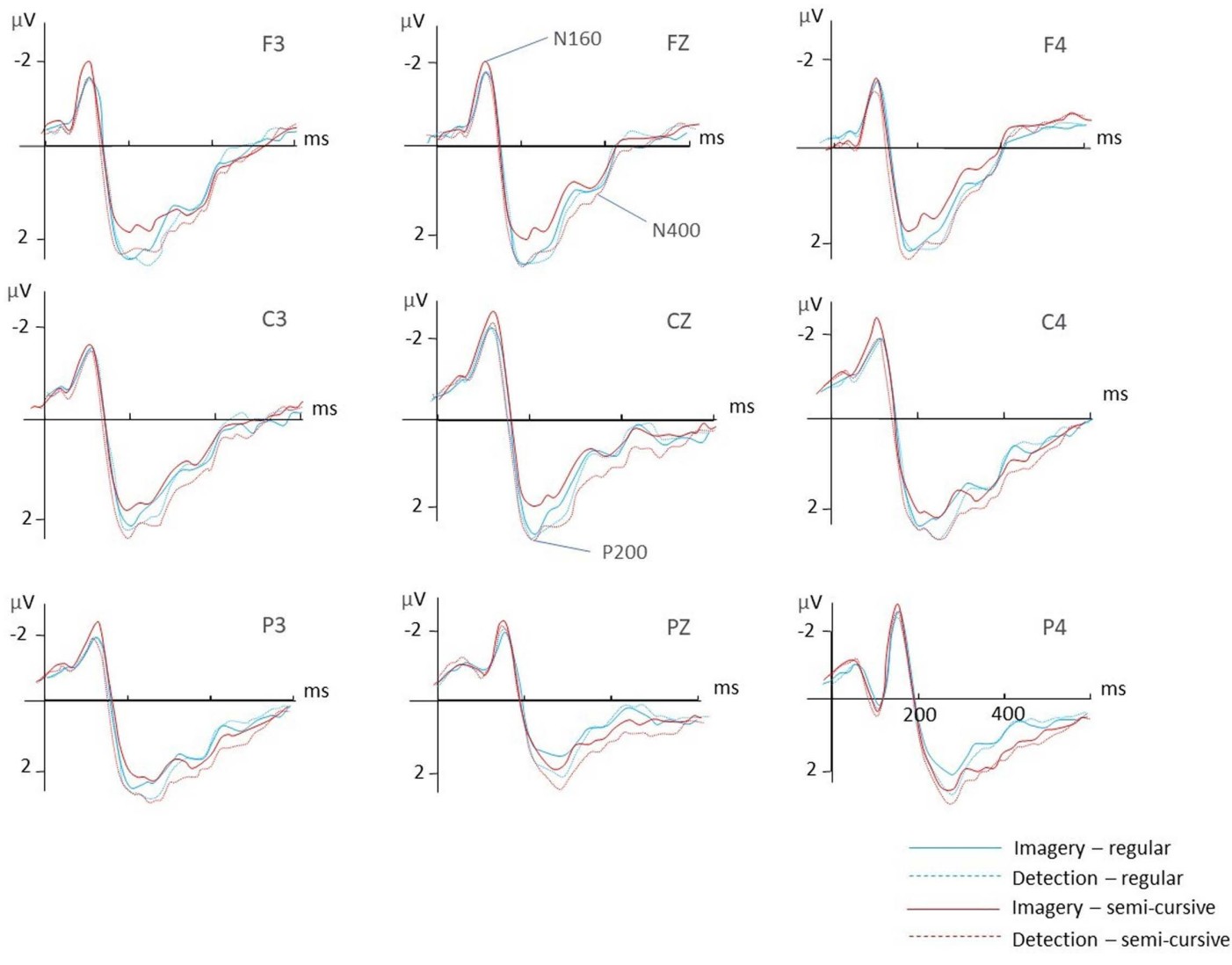

**Fig 2. Grand averages event-related potentials (ERPs) were recorded at nine electrodes across frontal, central and parietal regions during the perception/detection task.** The waveforms are superimposed for the two conditions with different familiarity levels. The blue solid line represents the imagery condition involving familiar Chinese strokes while the blue dashed line represents the detection condition with familiar Chinese strokes. The red solid line represents the imagery condition with unfamiliar Chinese strokes. The red dashed line represents the detection condition with unfamiliar Chinese strokes.

increased as signified by the N160 component but such an increase was not obtained for the external-to-internal attention orientation of the unfamiliar stimuli as shown by the P200 component. More importantly, the findings revealed that the top-down imagery generation of unfamiliar stimulus reduced the mental effort of the contextual integration as reflected by the N400 component. The findings of this ERP study provide a new insight that the neural processes of contextual integration are not only mediated by the mental effort for external-to-internal attentional control but also by the top-down generation and rehearsal of internal representation. The adoption of Chinese strokes as the medium in this study also implies that the contextual integration could be mediated at the orthographic level which do not carry explicit semantics.

## 4.1. Stimulus familiarity affects attention orientation

The current study involved two styles of Chinese strokes, representing a different level of familiarity to our participants. The level of stimulus familiarity has a significant impact on attention orientation, with unfamiliar stimuli often requiring greater attentional resources and resulting in a different pattern of neural activity compared to familiar stimuli. The participants are supposed to be more familiar with the regular style Chinese character while unfamiliar with the semi-cursive style Chinese characters as all participants had little or no Chinese calligraphy experiences. The characters with semi-cursive style were regarded as novel to them. Results showed that the participants performed imagery and detection of unfamiliar semi-cursive strokes elicited a larger amplitude of the N160 component than the regular style. The N160 component is evoked when the participants selectively attend to a feature of a visual stimulus [54]. It is stipulated that unfamiliar visual patterns or words attract people's attention [55]. The onset of unfamiliar semi-cursive strokes in this study elicited a larger N160 component, suggesting that higher attention control was required to process unfamiliar sub-character level Chinese strokes. The participants directed their attention to stroke onsets due to the fact that the presented semi-cursive Chinese strokes did not fit with their existing internal representation. A study using pseudo-letters also found a similar result that higher attentional control was required to process unfamiliar pseudo-letters [34,36]. The P200 component has been linked to higher-order processes of attention and perception [40,56]. It has been found to be associated with attention orientation from the external to the internal environment [57,58]. In this study, less positive P200 in the frontal region was observed in imagery trials compared with detection trials viewing Chinese characters with unfamiliar semi-cursive style. In contrast, such discrepancy was not observed in trials viewing familiar regular-style Chinese which might indicate more novel stimuli require higher demand on the attention orientation process. This implies that the anticipatory visual image of the unfamiliar representations of the semi-circular characters would mediate the external-to-internal attention orientation.

## 4.2. Effects of stimulus familiarity on contextual integration

The level of stimulus familiarity was shown to mediate the integration of unfamiliar stimuli into internal representations. In the context of the semantics of lexicons, the N400 component was suggested to be associated with contextual integration representing the mental discrepancy between bottom-up stimulus and the internal representation in the long-term storage [4,40]. There was a study investigating the semantic context using one-character Chinese words as well [36]. It was found that semantically related Chinese words like 'shallow' and 'deep' yielded a less negative N400 than semantically unrelated Chinese words. However, the study perspective remained at the semantic level of the Chinese word. The current study used Chinese strokes in the semi-cursive style which does not carry explicit semantics to investigate of the neural processes relate to the context integration. It was found that strokes in semi-cursive style elicited a larger N400 amplitude in the participants who were not familiar with the writing style when compared to familiar regular style counterparts. A larger N400 amplitude has been revealed to signify a large discrepancy between the incoming stimuli with incongruent meaning with the existing internal representation [59,60]. The larger amplitude of the N400 component obtained in the participants when viewing semi-cursive style Chinese and improper sequence suggests that the internal representation of a semi-cursive Chinese stroke form has not been well established in the brain network, and a stronger mental effort for contextual integration is resulted. Less negative N400 component amplitude has been found when the discrepancy between anticipated and encountered information about aspects of meaning was smaller in a sentence [4,61]. This appears to be consistent with previous study findings. An earlier study on Chinese words also revealed a larger N400 component viewing stimuli less familiar to the Chinese-speaking participants. The participants were required to complete a delayed character-matching task with Chinese words. Participants needed to match probe characters with preceding fragments, radical-deleted or stroke-deleted [42,62]. Stroke-deleted fragments are less familiar to the participants resulting in a larger amplitude in the N400 component. It was reasoned that the participants in this study might mentally generated irrelevant or distorted images of semi-cursive Chinese, or activated the familiar regular style which requires more mental effort to

integrate external stimuli and reshape the internal representation. Another study on word recognition has examined the Chinese word processing of true words and pseudowords using both ERP measures and behavioral outcomes. The reaction time of pseudo words was found to be longer than that of true words, and the amplitude of the N400 component evoked by pseudo words was larger than true words [63]. This is in line with the hypothesis that additional mental effort is needed when the representation of unfamiliar stimuli is integrated into an internal representation. Levels of familiarity therefore mediate the contextual integration of unfamiliar stimuli into internal representation. The current study used Chinese strokes in semi-cursive and regular styles which represent different levels of familiarity to investigate the effect of context integration. The significant amplitude difference in the N400 component was obtained when participants viewed the unfamiliar semi-cursive style of Chinese strokes suggests that the contextual integration effect operates not only at the semantic level but also extends to orthographic processing.

### 4.3. Effect of visual imagery on context integration

Another aim of this study was to examine if context integration could be mediated by the top-down generation of the internal representation. The empirical data of this study showed that the effect of visual imagery was shown to be facilitative for context integration to integrate unfamiliar information into existing mental representations. The current study recorded the changes in the N400 component amplitude during participants' imagery of Chinese characters strokes. Chinese stroke sequence is stored as a long-term representation of individuals who are familiar with Chinese writing. While observing the appearance of strokes of a Chinese character, Chinese individuals would automatically generate an image of the upcoming strokes from long-term storage [64,65]. If there is any improper sequence of stroke appearance, the unfamiliar form of the character would require higher attention to process the incoming bottom-up stimuli, as reflected by a larger N160 amplitude. Further, the unfamiliar form would induce a discrepancy between bottom-up attention processing of unfamiliar form of characters and the top-down integration to long-term storage. This appears to lead to a larger effort of context integration as shown by increased N400 amplitude. On the other hand, the reduced amplitude of the N400 component during the mental generation of the visual image before the appearance of the stimuli indicate that the prior generation of the approximated internal representation appears to reduce the discrepancy between external and internal representations and the effort of context integration. As for the interaction effect of the stimulus familiarity and visual imagery, generating the image of Chinese strokes with familiar regular style was found to elicit a less negative N400 effect compared with control condition over the fronto-parietal areas (Fz, Cz & Pz) while generating the image of less familiar semi-cursive Chinese was revealed to elicit a more negative N400 effect compared with control detection condition. Qiu & Zhou's study [65] has found a similar effect while asking participants to perceive well-learned stroke sequences. It was found that native Chinese readers use their sequential knowledge to anticipate upcoming strokes in perceiving the writing of Chinese characters. Less negative N400 resulted when the participants perceived the proper stroke sequence, which matched with their anticipated internal representation. Therefore, the contextual integration effect does not merely occur at the word semantic level but also the form or style of the Chinese strokes level. Regular-style Chinese presented in the experiment are more consistent with the participants' stored representation in terms of the features of the next Chinese strokes. Less additional mental effort would be required to update and consolidate the internal representation. In contrast, additional mental effort is required in semi-cursive Chinese stroke since the internal representations related to semi-cursive forms have not been well established, and a larger effort on contextual integration is required for consolidation as shown by larger amplitude of the N400 component. The image generation of the approximated visual stimuli would require one to retrieve an internal representation of characters leading to less effortful contextual integration as reflected by the decreased negative N400 component. Perceiving the reality requires our attention to select meaningful patterns in sensory information, creating internal representations of words [4,66], visual objects, and faces [11,12] and scenes [14,15]. Contextual integration is associated with the level of familiarity therefore structuring the internal representation of words, faces, scenes, and so on.

The implications of this study findings let us have a better understanding of how the top-down control of attention and the generation of visual imagery could reshape and enrich our knowledge of the contexts. The process of contextual integration could apply to various visual or other modality stimuli without explicit semantics. The findings suggest that individuals expend greater cognitive effort when processing unfamiliar stimuli, requiring heightened attentional focus and additional mental resources to integrate new information into existing representations. Furthermore, the role of visual imagery in facilitating contextual integration underscores the importance of top-down cognitive mechanisms in processing novel stimuli. This research provides valuable insight into how familiarity shapes neural processing, offering directions for exploring attentional control and memory retrieval across different domains, such as language, object recognition, and complex scene interpretation. Future investigations could extend these findings to diverse populations and stimulus types, refining our understanding of the neural mechanisms underlying cognitive adaptability and learning.

## 5. Conclusion

This ERP study investigated the mediating effects of attentional control and visual imagery of different familiarity levels of Chinese strokes on the process of contextual integration. Also, we found that familiarity with incoming stimulus and related internal visual representations would mediate attentional demand (reflected by the N160 and P200 components) and contextual integration (reflected by the N400 component). We also found that visual imagery could pre-activate the generation of internal representation from long-term storage and facilitate the process of context integration. The findings may provide a new insight that the effort of contextual integration could be associated with both external-to-internal attentional control and internal representation generation from long-term storage across different visual stimuli. Besides, the N400 component could serve as another biomarker to indicate attention control and memory retrieval functions.

## Supporting information

**S1 File. The Supplementary Data for the N160 Component.**
(XLSX)

**S2 File. The Supplementary Data for the P200 Component.**
(XLSX)

**S3 File. The Supplementary Data for the N400 Component.**
(XLSX)

## Author contributions

**Conceptualization:** Sam Chi Chung Chan.

**Data curation:** Sam Chi Chung Chan.

**Formal analysis:** Sam Chi Chung Chan, Tom Chun Wai Tsoi.

**Methodology:** Sam Chi Chung Chan, Tom Chun Wai Tsoi.

**Supervision:** Sam Chi Chung Chan.

**Writing – original draft:** Sam Chi Chung Chan, Tom Chun Wai Tsoi.

**Writing – review & editing:** Sam Chi Chung Chan, Tom Chun Wai Tsoi.

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
