## [Decision Letter · Decision Letter 0]

19 Mar 2025

PONE-D-25-00635The Familiarity Effect of Chinese Stroke Stimulus and Imagery on Contextual Integration: Evidence from ERP CorrelatesPLOS ONE

Dear Dr. Chan,

Thank you for submitting your manuscript to PLOS ONE. Your manuscript has been reviewed by two experts in the field. Both of them agreed that the topic of the study is interesting, but they also raised some conceptual and methodological issues that need to be addressed. You can find their comments in this decision letter. We invite you to submit a revised version of the manuscript that addresses the points raised during the review process. In particular, please pay special attention to R1's comment about the small sample size. Please provide a power analysis to address this. In addition, please also state the number of items you have used in stimuli (currently mentioned in paradigm: 4 blocks of 16 trials per block. But it is unclear whether there were repetitions). Both pieces of information are important to evaluate the generalizability of your findings.

We look forward to receiving your revised manuscript.

Kind regards,

Yiu-Kei Tsang

Academic Editor

PLOS ONE

2. In the online submission form, you indicated that [Data cannot be shared publicly because of participants' confidentiality. Data are available from the Dpearmental Ethics Committee (contact via Dr Sam Chi Chung Chan) upon request.].

Reviewers' comments:

Reviewer's Responses to Questions

**Comments to the Author**

1. Is the manuscript technically sound, and do the data support the conclusions?

Reviewer #1: Yes

Reviewer #2: Partly

2. Has the statistical analysis been performed appropriately and rigorously? 

Reviewer #1: Yes

Reviewer #2: Yes

3. Have the authors made all data underlying the findings in their manuscript fully available?

Reviewer #1: Yes

Reviewer #2: Yes

4. Is the manuscript presented in an intelligible fashion and written in standard English?

Reviewer #1: Yes

Reviewer #2: Yes

5. Review Comments to the Author

Reviewer #1: The manuscript presents a technically sound study with a clear hypothesis and experimental design. The use of ERP components (N160, P200, N400) to investigate attentional control and contextual integration in processing familiar vs. unfamiliar Chinese strokes is well-grounded in prior literature. The tasks (imagery vs. detection) and stimuli (semi-cursive vs. regular styles) are appropriately designed to address the research questions. The study addresses an underexplored topic (sub-character-level contextual integration) using a novel paradigm with Chinese strokes. The integration of ERP components to dissociate attentional and semantic processes is methodologically robust.

The study offers valuable insights into contextual integration mechanisms but requires methodological refinements, statistical transparency, and language improvements to meet publication standards. Theoretically, the authors may discuss how findings extend beyond orthographic processing to general mechanisms of contextual integration (e.g., cross-modal comparisons).

The sample size (N=19) is relatively small for ERP studies, which may limit generalizability. While the results align with the hypotheses (e.g., larger N400 amplitudes for unfamiliar stimuli), the lack of behavioral data (e.g., accuracy, reaction times) to corroborate ERP findings weakens the conclusions.

The statistical analysis is generally appropriate, employing three-way repeated-measures ANOVA with Greenhouse-Geisser correction for sphericity violations. Post-hoc tests with Bonferroni adjustments are justified. However, critical details are missing:

1�The exact p-values for ANOVA results (e.g., p<0.005 for N160) are inconsistently reported.

2�Effect sizes (e.g., η²) are not provided, making it difficult to assess the magnitude of observed effects.

3�The rationale for selecting midline electrodes (Fz, Cz, Pz) over other regions is not explicitly stated, raising concerns about potential cherry-picking.

4�The manuscript does not address potential outliers or data normality assumptions.

These omissions reduce the rigor and transparency of the analysis. The author may want to justify the sample size (N=19) with a power analysis. Include demographic details (e.g., educational background, handedness) to contextualize findings. And also report accuracy and reaction times to validate ERP results and rule out task performance confounds.

The writing is generally clear but contains grammatical errors and awkward phrasing that hinder readability. Thorough language editing is required to meet PLOS ONE’s standards.

1�Typos: “heighted” → “heightened” (Abstract, Line 4); “abd” → “and” (Page 12, Line 69).

2�Inconsistent terminology: “semi-cursive” vs. “Shinsu” (Page 19, Line 202); “radial-omitted” → “radical-omitted” (Page 14, Line 131).

3�Ambiguous sentences: E.g., “The education level of all selected characters was up to primary six” (Page 19, Line 208) – clarify if this refers to character complexity or participant education.

4�Formatting issues: Missing spaces (e.g., “N400 300-500ms)” → “N400 (300-500 ms)”), inconsistent citation styles (e.g., “Kutas & Federmeier, 2011” vs. “Kutas et al., 2011”).

Reviewer #2: Reviewer Report for Manuscript PONE-D-25-00635

Title: The Familiarity Effect of Chinese Stroke Stimulus and Imagery on Contextual Integration: Evidence from ERP Correlates

General Comments

This manuscript investigates how familiarity with Chinese stroke stimuli influences attentional processes and contextual integration, using ERP components (N160, P200, and N400) as neural markers. The focus on sub-character (stroke-level) processing is innovative and has the potential to contribute to our understanding of orthographic processing in Chinese. However, the study’s significance is undermined by a lack of clarity in its core concept—"incongruency" at the stroke level—as well as insufficient theoretical grounding, undefined terms, and methodological ambiguities. Addressing these issues through minor revisions will enhance the manuscript’s rigor and impact.

Major Concerns

1. Unclear Definition of "Incongruency" at the Stroke Level

The central concept of "incongruency" at the sub-character (stroke) level is not clearly defined, leaving readers uncertain about what it represents and why it matters. Specifically:

• The authors do not specify whether the incongruency pertains to stroke sequences during production (e.g., writing) or recognition (e.g., reading/perceiving). For example, if unexpected stroke sequences are presented in playback, this could reflect a production issue (deviating from how strokes are typically written) rather than a recognition issue (deviating from how strokes are expected in perception). Alternatively, it could involve both, but this ambiguity need to be addressed. The N400, typically linked to unexpected patterns or meanings, is used here to index stroke-level incongruency, but its relevance depends on whether the authors are targeting recognition, production, or a combination.

• The study appears to assume that stroke-level information (e.g., sequence or order) is stored and actively used in character recognition, making "incongruency" at this level detectable and meaningful. However, this assumption is neither explicitly stated nor supported with evidence. Without this foundation, the premise of studying stroke-level incongruency lacks justification.

• The authors frequently mention "integration," but it’s unclear how stroke-level incongruency relates to this process. If integration is about recognition (e.g., combining stroke information into a character), the design should test recognition-specific effects. If it involves production, this should be clarified. The current ambiguity makes it difficult to assess how stroke-level processing fits into the broader orthographic processing framework.

Recommendations:

• Explicitly define "incongruency" and specify whether it relates to production, recognition, or both.

• State the assumption that stroke-level information is stored and used in recognition (or production), and provide literature support (e.g., studies on stroke sequence encoding, sub-character processing, or orthographic integration in Chinese).

• Clarify how stroke-level incongruency contributes to orthographic integration, aligning it with the study’s research question.

2. Undefined Key Terms

Terms like "attention," "selective attention," "orientation of attention," and "integration" are used frequently but not defined. For instance, does "integration" refer to combining strokes into characters, contextual understanding, or something else? Brief definitions upon first use would improve clarity.

3. Methodological concern

• The choice of midline sites (Fz, Cz, Pz) for ERP analysis is not explained. Why were these sites chosen for N160, P200, and N400 effects?

• Applying the N400—typically a semantic processing marker—to non-semantic stroke stimuli is less common. The authors need to provide precedent (e.g., N400 effects in orthographic or non-linguistic contexts) to support this approach.

Minor Concerns

• The assumption that semi-cursive strokes are unfamiliar to participants is untested. Evidence (e.g., a familiarity survey or participant background data) should confirm this, if this is difficult to achieve, author can consider providing support from literature

• The term "optimal balance" (related to attention) is vague. What does it mean, and how does it connect to the study’s goals?

General Recommendation

This study’s focus on sub-character (stroke-level) processing is innovative and has the potential to advance understanding of orthographic processing in Chinese. However, the manuscript would benefit from clearer definition of "incongruency" (production vs. recognition), further clarification of assumptions regarding stroke-level storage, and additional theoretical and methodological explanations. I recommend minor revision to address these points:

• Define "incongruency" and clarify whether it pertains to production, recognition, or both, linking it to orthographic integration.

• Justify the assumption that stroke-level information is stored and used, with literature support.

• Improve theoretical rationale for studying stroke-level incongruency.

• Define key terms (e.g., "attention," "integration").

• Justify electrode sites and N400 use, and detail the experimental paradigm.

• Verify stimulus familiarity assumptions (or with support from literature).

6. PLOS authors have the option to publish the peer review history of their article (what does this mean? ). If published, this will include your full peer review and any attached files.

**Do you want your identity to be public for this peer review?** For information about this choice, including consent withdrawal, please see our Privacy Policy .

Reviewer #1: No

Reviewer #2: No

---

## [Author Response · Author response to Decision Letter 1]

19 May 2025

Comments Responses

(Line numbers as shown in the Revised Manuscript with Track Changes

Editor’s Comment

Please provide a power analysis to address this. Thanks for the suggestion. Power analysis has been conducted and

Sample estimation is reported in the Materials and Methods section (Section 2.2). Tom ok

In addition, please also state the number of items you have used in stimuli (currently mentioned in paradigm: 4 blocks of 16 trials per block. But it is unclear whether there were repetitions). Both pieces of information are important to evaluate the generalizability of your findings.

Number of Chinese characters used is included in the paradigm section (Section 2.4, Line 315 - 317). Tom ok

Reviewer #1

1.1 The manuscript presents a technically sound study with a clear hypothesis and experimental design. The use of ERP components (N160, P200, N400) to investigate attentional control and contextual integration in processing familiar vs. unfamiliar Chinese strokes is well-grounded in prior literature. The tasks (imagery vs. detection) and stimuli (semi-cursive vs. regular styles) are appropriately designed to address the research questions. The study addresses an underexplored topic (sub-character-level contextual integration) using a novel paradigm with Chinese strokes. The integration of ERP components to dissociate attentional and semantic processes is methodologically robust.

Thanks for your valuable comments.

1.2 The study offers valuable insights into contextual integration mechanisms but requires methodological refinements, statistical transparency, and language improvements to meet publication standards.

Theoretically, the authors may discuss how findings extend beyond orthographic processing to general mechanisms of contextual integration (e.g., cross-modal comparisons). Thanks for the comments. The Section 2.2 Sample Size Estimation, and the statistical details are added in the section 3 of Result.

Generalization / implications are explained in the discussion section (Line 632 - 646)

The manuscript has been copyedited. editing

ok

generalization ok

1.3 The sample size (N=19) is relatively small for ERP studies, which may limit generalizability. While the results align with the hypotheses (e.g., larger N400 amplitudes for unfamiliar stimuli), the lack of behavioral data (e.g., accuracy, reaction times) to corroborate ERP findings weakens the conclusions.

Thanks for the comments. Sample estimation is included in the Materials and Methods section (Section 2.2).

Besides, the p-values reported in the results (Section 3.2 – 3.4) were relatively small, implying sufficient sample power. tom OK

The statistical analysis is generally appropriate, employing three-way repeated-measures ANOVA with Greenhouse-Geisser correction for sphericity violations. Post-hoc tests with Bonferroni adjustments are justified.

Thanks for the comments

However, critical details are missing:

1�The exact p-values for ANOVA results (e.g., p<0.005 for N160) are inconsistently reported.

Exact p-values are reported in the Section 3.2 – 3.4 of the Results section. tom OK

2�Effect sizes (e.g., η²) are not provided, making it difficult to assess the magnitude of observed effects.

The values of effect size (η²) are reported in Section 3.2 – 3.4 of the Results section. tom OK

3�The rationale for selecting midline electrodes (Fz, Cz, Pz) over other regions is not explicitly stated, raising concerns about potential cherry-picking.

The reasons to choose these three midline sites (Fz, Cz and Pz) are provided in Section 2.7 (Line 388 – 393). tom ok

4�The manuscript does not address potential outliers or data normality assumptions.

These omissions reduce the rigor and transparency of the analysis. The author may want to justify the sample size (N=19) with a power analysis. Include demographic details (e.g., educational background, handedness) to contextualize findings. Power analysis and Sample estimation is included in the Section 2.2 Sample Size Estimation.

Outlier analysis was added in Section 2.6 (Line 350 – 352). No outlier participants were identified

Offline preprocessing procedure was applied to reject outlying EEG data as described in Section 2.6 (Line 375 – 376).

Demographic details of the participants are included in Section 2.1 Participants as suggested (Line 257 – 260). tom (sample size estimation to be added)

And also report accuracy and reaction times to validate ERP results and rule out task performance confounds. The table of accuracy and reaction of the behavioral data during the experience (Table 1) are added as suggested (Line 428-431). The results are summarized in the main text (Section 3.1) by Tom ok

The writing is generally clear but contains grammatical errors and awkward phrasing that hinder readability. Thorough language editing is required to meet PLOS ONE’s standards.

1�Typos: “heighted” → “heightened” (Abstract, Line 4); “abd” → “and” (Page 12, Line 69).

2�Inconsistent terminology: “semi-cursive” vs. “Shinsu” (Page 19, Line 202); “radial-omitted” → “radical-omitted” (Page 14, Line 131).

3�Ambiguous sentences: E.g., “The education level of all selected characters was up to primary six” (Page 19, Line 208) – clarify if this refers to character complexity or participant education.

4�Formatting issues: Missing spaces (e.g., “N400 300-500ms)” → “N400 (300-500 ms)”), inconsistent citation styles (e.g., “Kutas & Federmeier, 2011” vs. “Kutas et al., 2011”).

The following revisions are made as suggested:

1�Typos: “heighted” has been revised to “heightened” (Abstract, Line 24) and “abd” has been revised to “and” (Line 100).

2� “Shinsu” is specified as the name of font used in the paradigm (Line 275)

“radial-omitted” has been amended to “radical-omitted” (Line 183).

The “education level” is replaced by the “character complexity” (Line 280)

“N400 300-500ms)” is changed to “N400 (300-500 ms)” (Line 34)

The citations are all standardized to be “Kutas & Federmeier” (Line 62, 158, 174, & 539) OK

Reviewer #2

This manuscript investigates how familiarity with Chinese stroke stimuli influences attentional processes and contextual integration, using ERP components (N160, P200, and N400) as neural markers. The focus on sub-character (stroke-level) processing is innovative and has the potential to contribute to our understanding of orthographic processing in Chinese. However, the study’s significance is undermined by a lack of clarity in its core concept—"incongruency" at the stroke level—as well as insufficient theoretical grounding, undefined terms, and methodological ambiguities. Addressing these issues through minor revisions will enhance the manuscript’s rigor and impact.

Thanks for the comments. Specific changes are made according to the recommendations in the next section

Major Concerns

1. Unclear Definition of "Incongruency" at the Stroke Level

The central concept of "incongruency" at the sub-character (stroke) level is not clearly defined, leaving readers uncertain about what it represents and why it matters. Specifically:

Thanks for the comments. Specific changes are made according to the recommendations in the next section

• The authors do not specify whether the incongruency pertains to stroke sequences during production (e.g., writing) or recognition (e.g., reading/perceiving). For example, if unexpected stroke sequences are presented in playback, this could reflect a production issue (deviating from how strokes are typically written) rather than a recognition issue (deviating from how strokes are expected in perception). Alternatively, it could involve both, but this ambiguity need to be addressed. The N400, typically linked to unexpected patterns or meanings, is used here to index stroke-level incongruency, but its relevance depends on whether the authors are targeting recognition, production, or a combination.

require further discussion

• The study appears to assume that stroke-level information (e.g., sequence or order) is stored and actively used in character recognition, making "incongruency" at this level detectable and meaningful. However, this assumption is neither explicitly stated nor supported with evidence. Without this foundation, the premise of studying stroke-level incongruency lacks justification.

• The authors frequently mention "integration," but it’s unclear how stroke-level incongruency relates to this process. If integration is about recognition (e.g., combining stroke information into a character), the design should test recognition-specific effects. If it involves production, this should be clarified. The current ambiguity makes it difficult to assess how stroke-level processing fits into the broader orthographic processing framework.

Recommendations:

• Explicitly define "incongruency" and specify whether it relates to production, recognition, or both.

Thanks for the recommendations.

The term “incongruency” is defined more clearly in terms of recognition

(Line 86 - 92) Sam

check line number / section number

• State the assumption that stroke-level information is stored and used in recognition (or production), and provide literature support (e.g., studies on stroke sequence encoding, sub-character processing, or orthographic integration in Chinese) The assumption that the familiarity of stroke-level is stored and used in recognition is described as suggested (Line 97 - 109) Sam

check line number / section number

• Clarify how stroke-level incongruency contributes to orthographic integration, aligning it with the study’s research question. How stroke-level incongruency would lead an increased effort for the contextual integration is explained more clearly (Line 102-111) Sam

check line number / section number

2. Undefined Key Terms

Terms like "attention," "selective attention," "orientation of attention," and "integration" are used frequently but not defined. For instance, does "integration" refer to combining strokes into characters, contextual understanding, or something else? Brief definitions upon first use would improve clarity.

Definitions of “selective attention” and “attention orientation” are provided in section 1.1“ (Line 126-132)

Contextual integration” are explained in Line 111 - 122. Sam

ok

3. Methodological concern

• The choice of midline sites (Fz, Cz, Pz) for ERP analysis is not explained. Why were these sites chosen for N160, P200, and N400 effects?

The reasons to choose the midline sites at Fz, Cz and Pz are provided in Section 2.7 (Line 388-393).

OK

• Applying the N400—typically a semantic processing marker—to non-semantic stroke stimuli is less common. The authors need to provide precedent (e.g., N400 effects in orthographic or non-linguistic contexts) to support this approach. The N400 has been found in the context of visual scenes, such as face and traffic light identification. (Section 1, Line 83-85)

Minor Concerns

• The assumption that semi-cursive strokes are unfamiliar to participants is untested. Evidence (e.g., a familiarity survey or participant background data) should confirm this, if this is difficult to achieve, author can consider providing support from literature

Inclusion criteria included asking participants whether they have prior training in Chinese calligraphy, especially in the semi-cursive style.

“(2) being native Chinese and being able to read Chinese; (3) not having prior formal training in Chinese calligraphy in semi-cursive style”. (Section 2.1, Line 250-256)

Sam/

Tom

• The term "optimal balance" (related to attention) is vague. What does it mean, and how does it connect to the study’s goals? “optimal balance” replaced by “effective”(line 148)

General Recommendation

This study’s focus on sub-character (stroke-level) processing is innovative and has the potential to advance understanding of orthographic processing in Chinese. However, the manuscript would benefit from clearer definition of "incongruency" (production vs. recognition), further clarification of assumptions regarding stroke-level storage, and additional theoretical and methodological explanations. I recommend minor revision to address these points:

Thanks for the comments.

• Define "incongruency" and clarify whether it pertains to production, recognition, or both, linking it to orthographic integration. The term “incongruency” is defined more clearly in terms of recognition

(Line 86 - 92)

• Justify the assumption that stroke-level information is stored and used, with literature support. The assumption that stroke-level is stored and used in recognition is described as suggested (Line 97-109)

• Improve theoretical rationale for studying stroke-level incongruency. The theoretical rationale for studying stroke-level congruency is explained (Line 210-227)

• Define key terms (e.g., "attention," "integration"). Definitions of “selective attention” and “attention orientation” are provided in section 1.1 (Line 126-132)

• Justify electrode sites and N400 use, and detail the experimental paradigm. The reasons to choose these three midline sites (Fz, Cz and Pz) are provided in Section 2.7 (Line 388-393).

• Verify stimulus familiarity assumptions (or with support from literature). The assumption on the stroke-level familiarity is described as suggested with literature support (Line 102-111)

---

## [Decision Letter · Decision Letter 1]

10 June 2025

The Familiarity Effect of Chinese Stroke Stimulus and Imagery on Contextual Integration: Evidence from ERP Correlates

PONE-D-25-00635R1

Dear Dr. Chan,

We’re pleased to inform you that your manuscript has been judged scientifically suitable for publication and will be formally accepted for publication once it meets all outstanding technical requirements.

Kind regards,

Yiu-Kei Tsang

Academic Editor

PLOS ONE

Additional Editor Comments (optional):

Reviewers' comments:

Reviewer's Responses to Questions

**Comments to the Author**

1. If the authors have adequately addressed your comments raised in a previous round of review and you feel that this manuscript is now acceptable for publication, you may indicate that here to bypass the “Comments to the Author” section, enter your conflict of interest statement in the “Confidential to Editor” section, and submit your "Accept" recommendation.

Reviewer #1: All comments have been addressed

Reviewer #2: All comments have been addressed

2. Is the manuscript technically sound, and do the data support the conclusions?

Reviewer #1: Yes

Reviewer #2: Yes

3. Has the statistical analysis been performed appropriately and rigorously? 

Reviewer #1: Yes

Reviewer #2: Yes

4. Have the authors made all data underlying the findings in their manuscript fully available?

Reviewer #1: Yes

Reviewer #2: Yes

5. Is the manuscript presented in an intelligible fashion and written in standard English?

Reviewer #1: Yes

Reviewer #2: Yes

6. Review Comments to the Author

Reviewer #1: The authors have adequately addressed the concerns raised in my previous review. The current version demonstrates significantly improved logical coherence and enhanced clarity of presentation, making the manuscript more accessible to readers. The revisions to the methodology and analytical sections have strengthened the academic rigor and scholarly standards of the work. The modifications effectively resolve the previously identified issues while maintaining the scientific integrity of the research. Based on these substantial improvements, I recommend this manuscript for acceptance.

Reviewer #2: Despite the potential technical issue of internal notes and comments from the authors' team appearing in the response, the content remains understandable, and the revised manuscript indicates that the authors have carefully considered and addressed all of my comments and suggestions. I have no further concerns and consider the manuscript suitable for acceptance and publication.

7. PLOS authors have the option to publish the peer review history of their article (what does this mean? ). If published, this will include your full peer review and any attached files.

**Do you want your identity to be public for this peer review?** For information about this choice, including consent withdrawal, please see our Privacy Policy .

Reviewer #1: No

Reviewer #2: No

---

## [Editor Report · Acceptance letter]

PONE-D-25-00635R1

PLOS ONE

Dear Dr. Chan,

I'm pleased to inform you that your manuscript has been deemed suitable for publication in PLOS ONE. Congratulations! Your manuscript is now being handed over to our production team.

Kind regards,

on behalf of

Dr. Yiu-Kei Tsang

Academic Editor

PLOS ONE